# The effectiveness of radial extracorporeal shock wave therapy vs transcutaneous electrical nerve stimulation in the management of upper limb spasticity in chronic-post stroke hemiplegia–A randomized controlled trial

Iresha Dilhari Senarath[1]*, Randika Dinesh Thalwathte[2], Manoji Pathirage[3], Senanayake A. M. Kularatne[3]

1 Faculty of Allied Health Sciences, Department of Physiotherapy, University of Peradeniya, Sri Lanka,
2 Information and Statistics Division, Rajarata University of Sri Lanka, Mininthale, Sri Lanka, 3 Faculty of Medicine, Department of Medicine, University of Peradeniya, Peradeniya, Sri Lanka

☯ These authors contributed equally to this work.
* dilharisenarath@ahs.pdn.ac.lk, dilsenarath7@gmil.com

## Abstract

### Background

Traditionally both rESWT and TENS are used in treating post-stroke upper limb spasticity over years and their effectiveness had been assessed disjointedly. However, these methods were not yet compared for superiority.

### Objectives

To compare rESWT vs TENS to assess their effectiveness in different parameters of stroke such as stroke type, gender, and the affected side.

### Methods

The experimental group was treated with rESWT application to the middle of the muscle belly of Teres major, Brachialis, Flexor carpi ulnaris, and Flexor digitorum profundus muscles using 1500 shots per muscle, frequency of 5Hz, energy of 0.030 mJ/mm. The TENS was applied to the same muscles in the control group using 100 Hz for 15 minutes. Assessments were taken at the baseline (T0), immediately after first application (T1), and at the end of four-week protocol (T2).

### Results

Patients 106 with a mean age of 63.87±7.052 years were equally divided into rESWT (53) and TENS (53) groups including 62 males, 44 females, 74 ischemic, 32 hemorrhagic, affecting 68 right, and 38 left. Statistical analysis has revealed significant differences at T1 and T2

**Data Availability Statement:** All relevant data are within the Supporting Information files.

**Funding:** The author(s) received no specific funding for this work.

**Competing interests:** The authors have declared that no competing interests exist.

in both groups. But at T2 compared to T0; the rESWT group has reduced spasticity 4.8 times (95% CI 1.956 to 2.195) while TENS reduced by 2.6 times (95% CI 1.351 to 1.668), improved voluntary control by 3.9 times (95% CI 2.314 to 2.667) and it was 3.2 times (95% CI 1.829 to 2.171) in TENS group. Improvement of the hand functions of the rESWT group was 3.8 times in FMA-UL (95% CI 19.549 to 22.602) and 5.5 times in ARAT (95% CI 22.453 to 24.792) while thrice (95% CI 14.587 to 17.488) and 4.1 times (95% CI 16.019 to 18.283) in TENS group respectively.

## Conclusion

The rESWT modality is superior compared to the TENS modality for treating chronic post-stroke spastic upper limb.

## Introduction

Stroke is recognized as the second leading cause of death globally and a significant cause of long-term neurological disability in adults. It leaves almost 50% of its survivors disabled concerning arm-hand performance, often for the rest of their lives [1, 2]. Six months after the stroke episode, a considerable number (25–53%) of people who survive are dependent on at least one daily living activity which frequently comprises the use of unilateral or bilateral upper limb movement [3]. Post-stroke spasticity is estimated to affect up to 43% of stroke survivors which delays the patients' rehabilitation, specifically functional recovery [4]. It can be seen already in the first week after stroke onset [5] and develops in 14–44% of patients within twelve months after stroke [6]. The most frequent pattern of arm spasticity is shoulder internal rotation and adduction together with elbow flexion, wrist flexion, and flexion of the fingers [7, 8].

The modality TENS has been recommended for spasticity management for many years due to its low cost, the comfort of use, and absence of adverse reactions [9]. Some studies recently reported that shock wave therapy or radial extracorporeal shock wave therapy (rESWT) is a safe, noninvasive, substitute treatment for spasticity [10]. Many previous studies have been conducted to identify the effectiveness of rESWT and various electrical stimulation modalities separately, but the effectiveness between them was not yet compared. And also, its effects on the variables such as stroke type, gender, and affected side were not yet analyzed. Therefore, the aim of this study was to compare the rESWT versus TENS modalities on chronic post-stroke spastic upper limb management and to identify differences based on stroke type, gender, and affected side. The rESWT and TENS are leading modalities for spasticity reduction.

## Methodology

### Study design and patient recruitment

This study was designed as a prospective, randomized, comparative, single-blind study. Patients were recruited based on the 'Peradeniya Stroke Registry (PSR)', the register for acute stroke admissions to the Teaching Hospital, Peradeniya, Sri Lanka. Five hundred and forty-four (544) patients diagnosed with their first episodes of stroke and registered from January 2017 to the end of June 2019 were considered for this study. They were detected by a stroke for more than six months at the time of data collection. Therefore, they are considered chronic stroke patients. As the first step, 54 patients who were given thrombolysis, 22 deaths, and eight (8) patients who were diagnosed with both stroke types or non-mentioned diagnoses were

excluded from the PSR. Remained total number 460 were contacted by the principal investigator to sort out the eligibility (inclusion and exclusion criteria) and their consent for this study. Among them, five deaths were recorded and 268 had better recovery. Filtered 187 chronic stroke patients with spasticity were undergone a randomization process to allocate to treatment and control groups.

## Participants

Subjects diagnosed as stroke by a physician either ischemic or hemorrhagic, both male and female patients aged between 40 to 70 years, subjects with the first-ever stroke for more than six months resulting in hemiplegia and who are able and willing to participate in the four-week study and to sign the consent form were included in this study. Thrombosis type of stroke patients, persons with reflex sympathetic dystrophy/ severe shoulder subluxation/ shoulder dislocation/ contracture in the affected UL/ fixed deformity of hand/ shoulder hand syndrome, patients who have received Botox injection or acupuncture within the past six months to the affected UL, patients with complete sensory loss of upper limb, an unstable medical condition like severe hypertension, convulsion, behavior problems, and visually impaired patients were excluded from the study. Data were collected for one year and the three-month period from January 2020 to April 2021.

## Sampling method

To reduce a type II error and increase the power, a preliminary power analysis using G*power 3.1.9.7 computer program, based on the test family: t-tests, statistical test: means: Wilcoxon-Mann-Whitney test (two groups); power $(1-\beta) = 0.85$; $\alpha = 0.05$; effect size = 0.5 were used to calculate the sample size. It indicated that a total sample of 108 people would be needed [11].

**Data collection.** Patients were asked to be present, and the study was taken place at the Patients' Service Unit, Department of Physiotherapy, Faculty of Allied Health Sciences, University of Peradeniya, Sri Lanka. Before being assigned to one of two treatment groups, all subjects received a description of the study. After the participants had signed the informed written consent, they were randomly assigned to the rESWT treatment group or the TENS treatment group. The experimental group was treated with rESWT and the control group was treated with TENS. All the subjects were gone through examination criteria to assess the spasticity level and the functional ability level of their affected upper limb. The modified Ashworth scale (MAS) was used to assess the spasticity grading, and voluntary control of the affected hand was assessed with voluntary control grading (VCG). Fugl-Meyer motor assessment scale for upper limb (FMA-UL) and action research arm test (ARAT) were used to assess the functional ability of the affected upper limb. Subjects were not exposed unnecessarily throughout the clinical examination.

## Randomization

The randomization was done in different stages. First, the study sample of 108 subjects was selected by generating random numbers from filtered 187 chronic stroke patients. Then, the patients were divided into two groups as ischemic stroke and (76) and hemorrhagic stroke (32).

The randomization process for the ischemic group or hemorrhagic group was followed separately. Seventy-six ischemic patients were randomly divided into 19 groups containing four patients per each and 32 hemorrhagic patients were randomly divided into eight groups containing four patients per each. Then treatment appointments were given as one group for one

day (4 ischemic patients or 4 hemorrhagic patients per day). The first ischemic patient or the first hemorrhagic patient of the group was selected to either experimental or control group by 'Envelop method' (same-sized and same-colored two envelops) [12]. There were two pieces of paper, one is written as '1' and the other as '2'. Each envelope contained only one piece of paper. The patient selected either '1' or '2', but he/ she doesn't know what is meant by '1' or '2'. Only the principal investigator knows '1' is indicated for 'Experimental group' and '2' is indicated for 'Control group'. If the first member of the group was selected to the experimental group, the second member was included in the control group, then the third for experimental and forth for control group. Same treatment selection procedure was followed for both the stroke groups. The second co-author generate the random allocation sequence, enrolled participants, and assigned participants to the interventions. Only the participants were blinded to the intervention type.

Ethical clearance (2019/EC/35) for this study was obtained from Ethics Review Committee, Faculty of Medicine, University of Peradeniya, and it was registered under Sri Lanka Clinical trial Registry with the registration number of SLCTR/2019/040 to conduct as a clinical trial.

### Procedure

Initially, all the grading of the spasticity level, voluntary motor control, hand functions, and other measurements and data were gained through the assessment and clinical examination. If the consented subject belongs to the control group, they received TENS, and the experimental group received rESWT. All the participants received either TENS or rESWT once a week for four weeks continuously according to the group to which he or she was assigned. Assessments of MAS, VCG, FMA-UL, and ARAT were taken at the baseline (prior to application of the modality-T0), just after the first application (T1), and at the end of the four-week treatment protocol (T2).

**Application of TENS.**   The patient was in a seated position with the affected arm may or/ may not rest on a pillow. TENS was applied over Teres major, Brachialis, Flexor carpi ulnaris, and Flexor digitorum profundus muscles in the frequency of 100 Hz for fifteen minutes. The passive electrode was placed at the cervical region (C7) and the active electrode was placed at the middle of the muscle belly.

**Application of rESWT.**   The patient was in a seated position with the affected arm may or/ may not rest on a pillow. rESWT was applied to the middle of the muscle belly, 1500 shots per muscle, frequency of 5Hz, the energy of 0.030 mJ/mm, over the above-mentioned muscles. The handle/ wand of the therapy unit was perpendicular to the skin.

There were no adverse effects or harms reported from application of any modality.

### Data entry and statistical analysis

All the collected data were fed into a Microsoft Excel sheet and statistical analysis was conducted using SPSS 22. Paired sample t-test (*p*-value) was used to analyze the statistical significance of mean values of the continuous variables; grading for MAS and VCG, scores for FMA-UL and ARAT. The mean differences in the outcome variables were calculated for the experiment and control groups' pre and post-treatment. Comparisons of the outcome variables were conducted among T0 & T1, T0 & T2, and T1 & T2.

### Results

This study included 108 patients but two of them could not complete the four-week treatment protocol; hence they were excluded from the survey resulting in 106 patients. They were between ages 40–70 years with a mean age of 63.87±7.052 years comprising 62 (58.5%) males

**Table 1. Baseline characteristics.**

| Treatment group | Age (Mean-yrs) | MAS (Mean/ SD) | VCG (Mean/ SD) | FMA-UL (Mean/ SD) | ARAT (Mean/SD) |
|---|---|---|---|---|---|
| **Treatment option** | | | | | |
| rESWT (53) | 63.32 | 4.09 (0.74) | 1.85 (0.72) | 20.23 (7.77) | 7.58 (5.38) |
| TENS (53) | 64.42 | 4.17 (0.75) | 1.66 (0.81) | 17.00 (6.98) | 6.53 (5.79) |
| **Stroke type** | | | | | |
| 1-IS (37) | 63.32 | 4.16 (0.64) | 1.76 (0.60) | 20.73 (8.58) | 6.92 (5.10) |
| 1-HS (16) | 63.31 | 3.94 (0.93) | 2.06 (0.93) | 19.06 (5.51) | 9.13 (5.87) |
| 2-IS (37) | 63.68 | 4.16 (0.69) | 1.68 (0.67) | 17.22 (6.99) | 6.24 (5.65) |
| 2-HS (16) | 66.13 | 4.19 (0.91) | 1.63 (1.09) | 16.50 (7.17) | 7.19 (6.24) |
| **Gender** | | | | | |
| 1-M (28) | 63.07 | 4.11 (0.68) | 1.86 (0.65) | 21.32 (8.11) | 7.79 (5.40) |
| 1-F (25) | 63.60 | 4.08 (0.81) | 1.84 (0.80) | 19.00 (7.34) | 7.36 (5.47) |
| 2-M (19) | 63.79 | 4.12 (0.77) | 1.74 (0.79) | 17.74 (7.55) | 6.85 (5.87) |
| 2-F (34) | 65.53 | 4.26 (0.73) | 1.53 (0.84) | 15.68 (5.80) | 5.95 (5.75) |
| **Affected side** | | | | | |
| 1-Rt (35) | 64.17 | 4.00 (0.68) | 1.91 (0.74) | 20.31 (7.45) | 8.29 (5.22) |
| 1-Lt (18) | 61.67 | 4.28 (0.82) | 1.72 (0.82) | 20.06 (8.60) | 6.22 (5.60) |
| 2-Rt (33) | 63.79 | 4.21 (0.74) | 1.67 (0.78) | 16.88 (6.38) | 6.09 (4.97) |
| 2-Lt (20) | 65.45 | 4.10 (0.79) | 1.65 (0.87) | 17.20 (8.07) | 7.25 (7.02) |

rESWT-radial Extracorporeal shock wave therapy, TENS-Transcutaneous electrical nerve stimulation, 1-rESWT, 2-TENS, MAS-Modified Ashworth Scale, VCG-voluntary control grading, FMA-UL-Fugl Mayer assessment of upper limb, ARAT-Action research arm test, IS-Ischemic stroke, HS-Hemorrhagic stroke, M-Male, F-Female, Rt-Right, Lt-Left, SD-standard deviation

and 44 (41.5%) females. The affected part was the right side upper limb of 68 (64.2%) and left side of 38 (35.8%). Of them, 74 (69.8%) of ischemic stroke patients and 32 hemorrhagic patients (30.2%) and were assigned to two treatment groups as experimental (rESWT: 53 (50%)) and control (TENS: 53 (50%)), resulting in four major patients treatment groups: ischemic rESWT: 37 (34.9%), ischemic TENS: 37 (34.9%), hemorrhagic rESWT: 16 (15.1%) and, hemorrhagic TENS: 16 (15.1%). The baseline characteristics of the participants are included in Table 1.

Sixty-four out of 106 subjects (60.4%) reported having medical co-morbidities. Among those 64 (100% co-morbidities) the following conditions were testified as having alone or as a combination; 53 had hypertension (HTN-82.81%), 40 had Diabetes Mellitus (DM-62.5%), 12 had ischemic heart disease (IHD-18.75%), three had chronic kidney disease (CKD-4.68%), one had atrial fibrillation (AF-1.56%), and one had dyslipidemia (DLP-1.56%) previously diagnosed. Among those 64 subjects, 16 were presented with HTN alone (25%), 11 with HTN and DM (17.18%), 10 with HTN, DM and, DLP (15.62%), seven with DM alone (10.93%) and, five subjects presented with HTN, DM, and IHD (7.81%) combinations. These included patients who were taking medications only for their present co-morbidities. They were not under any medications to treat spasticity at the stage of patient recruitment and they didn't take any antispastic drugs throughout the treatment protocol. They were only treated with either rESWT or TENS for spasticity management.

## Spasticity grading: Modified Ashworth Scale (MAS)

Percentages of the mean differences were calculated between time points, then the main rESWT and TENS groups and subgroup measurements were compared. Spasticity was reduced in both the groups at T1 and T2, but the improvement was larger in the rESWT group

**Table 2. Spasticity grading by Modified Ashworth Scale (MAS); mean values and test results of paired-sample t-test.**

| Treat. group | Mean (SD) | | | | T0 & T1 | | | | | T0 & T2 | | | | | T1 & T2 | | | | |
|---|---|---|---|---|---|---|---|---|---|---|---|---|---|---|---|---|---|---|---|
| | T0 | T1 | T2 | Mean dif % | Corr. | Sig. | p-value | Effect size | Mean dif % | Corr. | Sig. | p-value | Effect size | Mean dif % | Corr. | Sig. | p-value | Effect size | |
| rESWT | 4.09 (0.74) | 3.30 (0.99) | 2.02 (0.77) | 13.21 | 0.929 | 0.000 | 0.000 | 1.9 | 34.59 | 0.838 | 0.000 | 0.000 | 4.8 | 21.38 | 0.897 | 0.000 | 0.000 | 2.8 | |
| TENS | 4.17 (0.75) | 3.94 (0.84) | 2.66 (0.78) | 3.77 | 0.865 | 0.000 | 0.000 | 0.5 | 24.84 | 0.719 | 0.000 | 0.000 | 2.6 | 21.7 | 0.817 | 0.000 | 0.000 | 2.6 | |
| 1-IS | 4.16 (0.65) | 3.46 (0.99) | 2.00 (0.71) | 11.71 | 0.924 | 0.000 | 0.000 1.5 | | 32.43 | 0.887 0.000 | | 0.000 5.9 | | 20.72 | 0.920 0.000 | | 0.000 2.8 | | |
| 1-HS | 3.94 (0.93) | 2.94 (0.93) | 1.56 (0.73) | 16.66 | * | * | * | * | 39.58 | 0.845 | 0.000 | 0.000 | 4.7 | 22.91 | 0.845 | 0.000 | 0.000 | 2.7 | |
| 2-IS | 4.16 (0.68) | 4.00 (0.81) | 2.78 (0.75) | 2.7 | 0.890 | 0.000 | 0.000 | 0.4 | 22.52 | 0.716 | 0.000 | 0.000 | 2.5 | 19.81 | 0.816 | 0.000 | 0.000 | 2.5 | |
| 2-HS | 4.19 (0.91) | 3.81 (0.91) | 2.38 (0.81) | 6.25 | 0.849 | 0.000 | 0.000 | 0.7 | 30.2 | 0.806 | 0.000 | 0.000 | 3.3 | 23.96 | 0.829 | 0.000 | 0.000 | 2.8 | |
| 1-M | 4.11 (0.68) | 3.29 (0.90) | 2.00 (0.67) | 13.69 | 0913 | 0.000 | 0.000 | 2.1 | 35.11 | 0.811 | 0.000 | 0.000 | 5 | 21.43 | 0.867 | 0.000 | 0.000 | 2.8 | |
| 1-F | 4.08 (0.81) | 3.32 (1.11) | 2.04 (089) | 12.66 | 0.943 | 0.000 | 0.000 | 1.7 | 34 | 0.861 | 0.000 | 0.000 | 4.5 | 21.33 | 0.918 | 0.000 | 0.000 | 2.8 | |
| 2-M | 4.12 (0.77) | 3.91 (0.83) | 2.62 (0.85) | 3.43 | 0.871 | 0.000 | 0.000 | 0.5 | 24.5 | 0.763 | 0.000 | 0.000 | 2.6 | 21.07 | 0.849 | 0.000 | 0.000 | 2.8 | |
| 2-F | 4.26 (0.73) | 4.00 (0.88) | 2.74 (0.65) | 4.38 | 0.859 | 0.000 | 0.000 | 0.6 | 25.43 | 0.616 | 0.000 | 0.000 | 2.5 | 21.05 | 0.771 | 0.000 | 0.000 | 2.2 | |
| 1-Rt | 4.00 (0.68) | 3.14 (0.88) | 1.91 (0.74) | 14.28 | 0.926 | 0.000 | 0.000 | 2.4 | 34.76 | 0.808 | 0.000 | 0.000 | 4.7 | 20.47 | 0.875 | 0.000 | 0.000 | 2.8 | |
| 1-Lt | 4.28 (0.82) | 3.61 (1.14) | 2.22 (0.81) | 11.11 | 0.929 | 0.000 | 0.000 | 1.4 | 34.26 | 0.871 | 0.000 | 0.000 | 5 | 23.15 | 0.925 | 0.000 | 0.000 | 2.7 | |
| 2-Rt | 4.21 (0.74) | 4.03 (0.81) | 2.67 (0.73) | 3.03 | 0.876 | 0.000 | 0.000 | 0.5 | 25.75 | 0.708 | 0.000 | 0.000 | 2.7 | 22.72 | 0.804 | 0.000 | 0.000 | 2.7 | |
| 2-Lt | 4.10 (0.79) | 3.80 (0.89) | 2.65 (0.87) | 5 | 0.851 | 0.000 | 0.000 | 0.6 | 23.3 | 0.740 | 0.000 | 0.000 | 2.4 | 18.3 | 0.847 | 0.000 | 0.000 | 2.3 | |

*The correlation and t cannot be computed because the standard error of the difference is 0.

IS-Ischemic stroke, HS-Hemorrhagic stroke, M-Male, F-Female, Rt-Right, Lt-Left, T0-baseline measures, T1-Immediate measures after 1st treatment, T2-Measures at the end of the four-week session, SD-standard deviation, Dif-difference, Corr.-Correlation, Sig.-Significance

with 1.9 times at T0 & T1 (95% CI 0.680 to 0.905), 4.8 times at T0 & T2 (95% CI 1.956 to 2.195), and 2.8 times at T1 & T2 (95% CI 1.158 to 1.408) compared to TENS group. In sub-group analysis, hemorrhagic patients have shown comparatively higher grades than ischemic patients at T1 and T2 compared to the baseline. In the rESWT group, males got higher grades compared to females and females scored high compared to males in the TENS group. In rESWT, 1-Rt has shown higher grades compared to the 1-Lt group. Strong positive significant linear correlations ($r > 0.5$, $p < 0.001$) were identified in time pairs in both groups (Table 2).

## Voluntary control of the upper limb: Voluntary control grading (VCG)

Baseline measurements are nearly the same in the main rESWT and TENS groups but all other measurements in different time points were comparatively highly improved in the rESWT group than TENS group; 1.6 times at T0 & T1 (95% CI 0.613 to 0.859), 3.9 times at T0 & T2 (95% CI 2.314 to 2.667), and thrice at T1 & T2 (95% CI 1.593 to 1.916). At the end of the four-week session (T2), comparatively and significantly more improvement was seen in the rESWT group. Among the subgroups of stroke types, hemorrhagic patients have shown significantly

**Table 3. Grading of the voluntary control of the upper limb by Voluntary Control Grading scale (VCG); mean values and test results of paired-sample t-test.**

| Treat. group | Mean (SD) | | | | T0 & T1 | | | | | T0 & T2 | | | | | T1 & T2 | | | | |
|---|---|---|---|---|---|---|---|---|---|---|---|---|---|---|---|---|---|---|---|
| | T0 | T1 | T2 | Mean dif % | Corr. | Sig. | p-value | Effect size | Mean dif % | Corr. | Sig. | p-value | Effect size | Mean dif % | Corr. | Sig. | p-value | Effect size |
| rESWT | 1.85 (0.72) | 2.58 (0.95) | 4.34 (0.76) | 10.51 | 0.894 | 0.000 | 0.000 | 1.6 | 35.58 | 0.626 | 0.000 | 0.000 | 3.9 | 25.1 | 0.787 | 0.000 | 0.000 | 3 |
| TENS | 1.66 (0.81) | 2.04 (0.94) | 3.66 (0.73) | 5.4 | 0.854 | 0.000 | 0.000 | 0.7 | 28.57 | 0.679 | 0.000 | 0.000 | 3.2 | 23.18 | 0.773 | 0.000 | 0.000 | 2.7 |
| 1-IS | 1.76 (0.59) | 2.41 (0.89) | 4.24 (0.76) | 9.26 | 0.865 | 0.000 | 0.000 | 1.3 | 35.52 | 0.685 | 0.000 | 0.000 | 4.4 | 26.25 | 0.830 | 0.000 | 0.000 | 3.6 |
| 1-HS | 2.06 (0.93) | 3.00 (0.96) | 4.56 (0.73) | 13.4 | 0.966 | 0.000 | 0.000 | 3.75 | 35.71 | 0.537 | 0.000 | 0.000 | 3 | 22.32 | 0.664 | 0.000 | 0.000 | 2.1 |
| 2-IS | 1.68 (0.67) | 1.97 (0.86) | 3.68 (0.67) | 4.24 | 0.848 | 0.000 | 0.000 | 0.6 | 28.57 | 0.690 | 0.000 | 0.000 | 3.8 | 24.32 | 0.752 | 0.000 | 0.000 | 3 |
| 2-HS | 1.63 (1.09) | 2.19 (1.11) | 3.63 (0.88) | 8.03 | 0.891 | 0.000 | 0.000 | 1.1 | 28.57 | 0.675 | 0.000 | 0.000 | 2.4 | 20.53 | 0.824 | 0.000 | 0.000 | 2.3 |
| 1-M | 1.86 (0.65) | 2.68 (0.86) | 4.39 (0.74) | 11.73 | 0.905 | 0.000 | 0.000 | 2.1 | 36.22 | 0.585 | 0.000 | 0.000 | 4 | 24.49 | 0.730 | 0.000 | 0.000 | 2.8 |
| 1-F | 1.84 (0.80) | 2.48 (1.04) | 4.28 (0.79) | 9.14 | 0.893 | 0.000 | 0.000 | 1.3 | 34.85 | 0666 | 0.000 | 0.000 | 3.7 | 25.71 | 0.838 | 0.000 | 0.000 | 3.1 |
| 2-M | 1.74 (0.79) | 2.09 (0.90) | 3.74 (0.66) | 5.04 | 0.843 | 0.000 | 0.000 | 0.7 | 28.57 | 0.669 | 0.000 | 0.000 | 3.3 | 23.52 | 0.748 | 0.000 | 0.000 | 2.7 |
| 2-F | 1.53 (0.84) | 1.95 (1.02) | 3.53 (0.84) | 6.01 | 0871 | 0.000 | 0.000 | 0.8 | 28.57 | 0.686 | 0.000 | 0.000 | 3 | 22.55 | 0.806 | 0.000 | 0.000 | 2.6 |
| 1-Rt | 1.91 (0.66) | 2.71 (0.86) | 4.37 (0.77) | 11.42 | 0.890 | 0.000 | 0.000 | 2 | 35.1 | 0.587 | 0.000 | 0.000 | 3.7 | 23.67 | 0.787 | 0.000 | 0.000 | 3 |
| 1-Lt | 1.72 (0.82) | 2.33 (1.08) | 4.28 (0.75) | 8.73 | 0.897 | 0.000 | 0.000 | 1.2 | 36.5 | 0.699 | 0.000 | 0.000 | 4.1 | 27.77 | 0817 | 0.000 | 0.000 | 3 |
| 2-Rt | 1.67 (0.78) | 2.03 (0.92) | 3.67 (0.69) | 5.19 | 0.847 | 0.000 | 0.000 | 0.7 | 28.57 | 0.658 | 0.000 | 0.000 | 3.2 | 23.37 | 0.754 | 0.000 | 0.000 | 2.7 |
| 2-Lt | 1.65 (0.87) | 2.05 (0.99) | 3.65 (0.81) | 5.71 | 0.864 | 0.000 | 0.000 | 0.8 | 28.57 | 0.707 | 0.000 | 0.000 | 3 | 22.85 | 0.801 | 0.000 | 0.000 | 2.7 |

IS-Ischemic stroke, HS-Hemorrhagic stroke, M-Male, F-Female, Rt-Right, Lt-Left, T0-baseline measures, T1-Immediate measures after 1[st] treatment, T2-Measures at the end of the four-week session, SD-standard deviation, Dif-difference, Corr.-Correlation, Sig.-Significance

more improvement than ischemic patients at the end of the four-week session compared to T0. Strong positive significant linear correlations ($r > 0.5$, $p < 0.001$) were identified in all-time pairs for rESWT, TENS groups, and the subgroups. The mean differences in the time pairs were statistically significant in all the groups (Table 3).

## Hand functions: Fugl-Meyer Assessment of Upper Limb (FMA-UL)

Hand functions were improved in both the groups at T1 and T2 but comparatively, scores at all the time points, including baseline, were higher in the rESWT group with the improvement of 2.3 times at T0 & T1 (95% CI 6.647 to 8.409), 3.8 times at T0 & T2 (95% CI 19.549 to 22.602), and thrice at T1 & T2 (95% CI 12.313 to 14.782). The main rESWT group and its subgroups have shown a comparatively more significant percentage of the mean difference than the main TENS group and its subgroups in every time pair. In subgroup analysis, hemorrhagic patients have shown comparatively higher scores than ischemic patients and 1-Rt compared to the 1-Lt group at T1 and T2 compared to the baseline. Strong positive significant linear correlations ($r > 0.5$, $p < 0.001$) were identified in time pairs of T0 & T1, T0 & T2, and T1 & T2 for rESWT, TENS groups and the subgroups except for T0 & T2 time pair in 1-HS and 2-F as it

**Table 4. Hand function scores by Fugl-Mayer assessment of upper limb (FMA-UL); mean values and test results of paired-sample t-test.**

| Treat. group | Mean (SD) | | | Mean dif % | T0 & T1 | | | Effect size | Mean dif % | T0 & T2 | | | Effect size | Mean dif % | T1 & T2 | | | Effect size |
|---|---|---|---|---|---|---|---|---|---|---|---|---|---|---|---|---|---|---|
| | T0 | T1 | T2 | | Corr. | Sig. | p-value | | | Corr. | Sig. | p-value | | | Corr. | Sig. | p-value | |
| rESWT | 20.23 (7.77) | 27.75 (8.68) | 41.30 (7.85) | 11.39 | 0.930 | 0.000 | 0.000 | 2.3 | 31.92 | 0.749 | 0.000 | 0.000 | 3.8 | 20.53 | 0.858 | 0.000 | 0.000 | 3 |
| TENS | 17.00 (6.98) | 22.02 (7.40) | 33.04 (7.21) | 7.60 | 0.869 | 0.000 | 0.000 | 1.5 | 24.30 | 0.726 | 0.000 | 0.000 | 3 | 16.69 | 0.863 | 0.000 | 0.000 | 2.8 |
| 1-IS | 20.73 (8.58) | 28.03 (9.52) | 41.51 (8.55) | 11.06 | 0.948 | 0.000 | 0.000 | 2.4 | 31.48 | 0.803 | 0.000 | 0.000 | 3.8 | 20.42 | 0.884 | 0.000 | 0.000 | 3 |
| 1-HS | 19.06 (5.51) | 27.13 (6.58) | 40.81 (6.14) | 12.23 | 0.844 | 0.000 | 0.000 | 2.3 | 32.95 | 0.472 | 0.065 | 0.000 | 3.6 | 20.73 | 0.729 | 0.001 | 0.000 | 2.9 |
| 2-IS | 17.22 (6.99) | 22.00 (7.52) | 32.92 (7.16) | 7.24 | 0.897 | 0.000 | 0.000 | 1.4 | 23.78 | 0.728 | 0.000 | 0.000 | 3 | 16.54 | 0.855 | 0.000 | 0.000 | 2.7 |
| 2-HS | 16.50 (7.17) | 22.06 (7.36) | 33.31 (7.55) | 8.42 | 0.807 | 0.000 | 0.000 | 1.2 | 25.46 | 0.728 | 0.001 | 0.000 | 3 | 17.04 | 0.883 | 0.000 | 0.000 | 3.1 |
| 1-M | 21.32 (8.11) | 29.07 (8.73) | 42.57 (8.16) | 11.74 | 0.936 | 0.000 | 0.000 | 2.5 | 32.19 | 0.722 | 0.000 | 0.000 | 3.5 | 20.45 | 0.842 | 0.000 | 0.000 | 2.8 |
| 1-F | 19.00 (7.34) | 26.28 (8.55) | 39.88 (7.39) | 11.03 | 0.922 | 0.000 | 0.000 | 2.1 | 31.63 | 0.770 | 0.000 | 0.000 | 4.2 | 20.60 | 0.870 | 0.000 | 0.000 | 3.2 |
| 2-M | 17.74 (7.55) | 22.79 (8.25) | 33.44 (7.89) | 7.65 | 0.902 | 0.000 | 0.000 | 1.4 | 23.78 | 0.769 | 0.000 | 0.000 | 3 | 16.13 | 0.885 | 0.000 | 0.000 | 2.7 |
| 2-F | 15.68 (5.80) | 20.63 (5.50) | 32.32 (5.93) | 7.5 | 0.746 | 0.000 | 0.000 | 1.2 | 25.21 | 0.583 | 0.009 | 0.000 | 3 | 17.71 | 0.788 | 0.000 | 0.000 | 3 |
| 1-Rt | 20.31 (7.45) | 28.31 (8.88) | 41.51 (8.61) | 12.12 | 0.915 | 0.000 | 0.000 | 2.2 | 32.12 | 0.741 | 0.000 | 0.000 | 3.6 | 20.00 | 0.868 | 0.000 | 0.000 | 3 |
| 1-Lt | 20.06 (8.60) | 26.67 (8.43) | 40.89 (6.33) | 10.01 | 0.977 | 0.000 | 0.000 | 3.5 | 31.56 | 0.820 | 0.000 | 0.000 | 4 | 21.54 | 0.853 | 0.000 | 0.000 | 3.2 |
| 2-Rt | 16.88 (6.38) | 22.00 (6.87) | 32.76 (5.91) | 7.75 | 0.838 | 0.000 | 0.000 | 1.3 | 24.06 | 0.712 | 0.000 | 0.000 | 3.4 | 16.30 | 0.868 | 0.000 | 0.000 | 3.1 |
| 2-Lt | 17.20 (8.07) | 22.05 (8.39) | 33.50 (9.12) | 7.34 | 0.904 | 0.000 | 0.000 | 1.3 | 24.69 | 0.744 | 0.000 | 0.000 | 2.6 | 17.34 | 0.872 | 0.000 | 0.000 | 2.5 |

IS-Ischemic stroke, HS-Hemorrhagic stroke, M-Male, F-Female, Rt-Right, Lt-Left, T0-baseline measures, T1-Immediate measures after 1st treatment, T2-Measures at the end of the four-week session, SD-standard deviation, Dif-difference, Corr.-Correlation, Sig.-Significance

has shown moderate positive nonlinear correlations ($r = 0.472$, $p = 0.065$ and $r = 0.583$, $p = 0.009$ respectively). The mean differences in the time pairs were statistically significant in all the groups (Table 4).

## Hand functions: Action Research Arm Test (ARAT)

Baseline measurements are nearly the same in the main rESWT and TENS groups but all other measurements in different time points were comparatively higher in the rESWT group than TENS group. The improvement was indicated as 2.4 times at T0 & T1 (95% CI 7.745 to 9.764), 5.5 times at T0 & T2 (95% CI 22.453 to 24.792), and thrice at T1 & T2 (95% CI 13.536 to 16.200) in the rESWT group. Percentages of the mean differences were comparatively and significantly higher in the hemorrhagic group than the ischemic group in both rESWT and TENS main groups, the 2-Lt, 2-M groups in T0 & T1, T0 & T2, T1 & T2 time pairs, and 1-F in T0 & T1 and T0 & T2 time pairs. Strong positive significant linear correlations ($r > 0.5$, $p < 0.001$) were identified in all-time pairs for rESWT, TENS groups and the subgroups except for T0 & T2 time pair in 1-HS as it has shown moderate positive nonlinear correlation ($r = 0.543$, $p = 0.0305$). The mean differences in the time pairs were statistically significant in all the groups (Table 5).

**Table 5. Hand function scores by Action Research Arm Test (ARAT); mean values and test results of paired-sample t-test.**

| Treat. group | Mean (SD) | | | | T0 & T1 | | | | | T0 & T2 | | | | | T1 & T2 | | | | |
|---|---|---|---|---|---|---|---|---|---|---|---|---|---|---|---|---|---|---|---|
| | T0 | T1 | T2 | Mean dif % | Corr. | Sig. | p-value | Effect size | Mean dif % | Corr. | Sig. | p-value | Effect size | Mean dif % | Corr. | Sig. | p-value | Effect size |
| rESWT | 7.58 (5.38) | 16.34 (7.74) | 31.21 (6.40) | 14.60 | 0.906 | 0.000 | 0.000 | 2.4 | 39.38 | 0.754 | 0.000 | 0.000 | 5.5 | 24.78 | 0.783 | 0.000 | 0.000 | 3 |
| TENS | 6.53 (5.79) | 11.06 (5.65) | 23.68 (2.67) | 7.51 | 0.899 | 0.000 | 0.000 | 1.7 | 28.58 | 0.771 | 0.000 | 0.000 | 4.1 | 21.06 | 0.840 | 0.000 | 0.000 | 3.7 |
| 1-IS | 6.92 (5.10) | 15.51 (7.95) | 29.70 (6.22) | 14.31 | 0.903 | 0.000 | 0.000 | 2.1 | 37.96 | 0.844 | 0.000 | 0.000 | 6.8 | 23.65 | 0.834 | 0.000 | 0.000 | 3.2 |
| 1-HS | 9.13 (5.87) | 18.25 (7.13) | 34.69 (5.53) | 15.20 | 0.924 | 0.000 | 0.000 | 3.2 | 42.60 | 0.543 | 0.030 | 0.000 | 4.7 | 27.40 | 0.644 | 0.007 | 0.000 | 3 |
| 2-IS | 6.24 (5.65) | 10.43 (5.72) | 22.62 (6.58) | 6.98 | 0.907 | 0.000 | 0.000 | 1.4 | 27.30 | 0.805 | 0.000 | 0.000 | 4.2 | 20.31 | 0.850 | 0.000 | 0.000 | 3.5 |
| 2-HS | 7.19 (6.24) | 12.44 (5.39) | 26.13 (4.80) | 8.75 | 0.897 | 0.000 | 0.000 | 1.9 | 31.56 | 0.758 | 0.001 | 0.000 | 4.6 | 22.81 | 0.812 | 0.000 | 0.000 | 4.3 |
| 1-M | 7.79 (5.40) | 16.11 (6.64) | 30.96 (5.28) | 13.86 | 0.956 | 0.000 | 0.000 | 3.8 | 38.61 | 0.798 | 0.000 | 0.000 | 6.8 | 24.75 | 0.848 | 0.000 | 0.000 | 4.2 |
| 1-F | 7.36 (5.47) | 16.60 (8.97) | 31.48 (7.57) | 15.40 | 0.886 | 0.000 | 0.000 | 1.9 | 40.20 | 0.745 | 0.000 | 0.000 | 4.7 | 24.80 | 0.745 | 0.000 | 0.000 | 2.4 |
| 2-M | 6.85 (5.87) | 11.65 (5.99) | 24.41 (6.71) | 8.00 | 0.908 | 0.000 | 0.000 | 1.8 | 29.26 | 0.794 | 0.000 | 0.000 | 4.2 | 21.26 | 0.836 | 0.000 | 0.000 | 3.4 |
| 2-F | 5.95 (5.75) | 9.95 (4.95) | 22.37 (5.30) | 6.66 | 0.892 | 0.000 | 0.000 | 1.5 | 27.36 | 0.729 | 0.000 | 0.000 | 4 | 20.70 | 0.837 | 0.000 | 0.000 | 4.2 |
| 1-Rt | 8.29 (5.22) | 16.69 (6.36) | 31.77 (6.09) | 14.00 | 0.962 | 0.000 | 0.000 | 4.2 | 39.13 | 0.786 | 0.000 | 0.000 | 6 | 25.13 | 0.859 | 0.000 | 0.000 | 4.5 |
| 1-Lt | 6.22 (5.60) | 15.67 (10.10) | 30.11 (7.02) | 15.75 | 0.891 | 0.000 | 0.000 | 1.6 | 39.81 | 0.692 | 0.001 | 0.000 | 4.6 | 24.06 | 0.720 | 0.001 | 0.000 | 2 |
| 2-Rt | 6.09 (4.97) | 10.45 (4.68) | 22.73 (5.76) | 7.26 | 0.877 | 0.000 | 0.000 | 1.8 | 27.73 | 0.700 | 0.000 | 0.000 | 4 | 20.46 | 0.777 | 0.000 | 0.000 | 3.4 |
| 2-Lt | 7.25 (7.02) | 12.00 (6.99) | 25.25 (6.89) | 7.91 | 0.916 | 0.000 | 0.000 | 1.6 | 30.00 | 0.847 | 0.000 | 0.000 | 4.7 | 22.08 | 0.904 | 0.000 | 0.000 | 4.3 |

IS-Ischemic stroke, HS-Hemorrhagic stroke, M-Male, F-Female, Rt-Right, Lt-Left, T0-baseline measures, T1-Immediate measures after 1st treatment, T2-Measures at the end of the four-week session, SD-standard deviation, Dif-difference, Corr.-Correlation, Sig.-Significance

## Discussion

This current study is the first of its kind to compare the effectiveness of radial extracorporeal shock wave therapy (rESWT) and transcutaneous electrical nerve stimulation (TENS) in treating the upper limb spasticity of patients with chronic post-stroke hemiplegia that analyzed the effectiveness among the stroke type, gender and affected side. The rESWT group has shown superior effects on spasticity reduction, voluntary control and hand functions over the TENS group, and hemorrhagic patients have better results compared to ischemic patients. No significant differences were identified according to gender and affected side.

Both groups have shown immediate after-effects at the end of the modality's first application in spasticity reduction and improving voluntary control of the upper limb but the rESWT group has shown larger improvement compared with TENS. Statistical analysis has shown that the mean differences in T0 & T1 pair in both the groups are significant ($p<0.05$). But the percentage of mean difference was high in the rESWT group. A study conducted in 2021 on post-stroke upper limb spasticity has found the same result as reduction of the spasticity level by 'one' MAS grade immediately after one application of rESWT [13]. It reflects that just after one application, rESWT has presented with positive results compared to the TENS application. Many more previous studies are supporting the immediate positive outcomes on spasticity

reduction after a single application of rESWT modality [10, 14–16], but no previous studies were found which assessed the immediate after-effects of a single application of TENS modality. This was the first study to assess immediate after-effects of TENS application on a spastic upper limb even though there were minor after effects.

Hand functions were improved in both the groups immediately after the first application of the modality, with statistical significance of the mean difference in FMA-UL and ARAT scales. But, comparatively, a higher percentage of mean difference was identified in the rESWT group than the TENS group. It was further clarified that a single application of both the rESWT and TENS modalities acted on improving the upper limb's motricity. Leng et al. (2021) [13], Li et al. (2016) [10], and Manganotti and Amelio (2005) [16] have found similar results with rESWT, and no previous evidence was found on TENS activity. This was the first study to identify the improved hand functions immediately after the first application of TENS.

At the end of the four-week treatment session (T2), both groups were presented with decreased spasticity grades, increased voluntary control grades, and improved hand function scores compared to the baseline and the first application. Still, the rESWT group has shown a significant difference from the TENS group. Previous studies have shown similar results at the end of their treatment period supporting the current findings; spasticity reduction and improvement in voluntary control grading and hand functions at the end of the four-week treatment protocol. The rESWT studies; two systematic reviews conducted by Poursaeed et al. (2021) [17] and Opara et al. (2021) [18], two RCTs (three-week, one session per week) conducted by Tabra et al. (2021) [19] and Wu et al. (2018) [20], and a case series (two-week, one session per week) conducted by Troncati et al. (2013) [21] have found that the reduction of spasticity level and enhancement of hand functions at the end of the treatment period which was assessed by MAS and FMA-UL tools. A similar outcome was found in TENS studies (RCTs) which were conducted by Karakuş et al. (2013; two-week) [22], Tilkici et al. (2017; three-week) [23], Yuzer et al. (2017; four-week) [24], Kim et al. (2013; four-week) [25], and de Kroon and IJzerman (2008; six-week protocol) [26].

## Limitations

The study was limited to four week period and two patients from the total sample of 108 were unable to participate due to the Lock-down in the country during the Covid-19 pandemic. Patients were recruited from the Teaching Hospital, Peradeniya, Sri Lanka, but due to the availability of electrical modalities they were asked to come to the Patient's Service Unit, Department of Physiotherapy, Faculty of Allied Health Sciences, University of Peradeniya. The treatment site was 3km distant from the patient recruitment site, which we can consider as a limitation of the study, as patients needed to travel one site to the other. Blinding and objective assessment of the patients were conducted by one another physiotherapist at the treatment site, if she is on a leave, another physiotherapist was asked to do that. It may has a minor affection on the study protocol. We were not able to work on prognostic factors of the chronic stroke upper limb disability which is another limitation of our study. And also we were unable to compensate the patients since there were no available funds to support this study.

## Conclusion

We found rESWT modality is superior to the TENS modality in treating chronic post-stroke spastic upper limb. Both the groups have shown improved hand functions from the first application at the end of the four-week protocol but greater scores in the rESWT group than the TENS group at T1 and T2. Hemorrhagic stroke patients have shown better grades than the ischemic stroke patients in spasticity reduction and improving voluntary control at the end of the four-week treatment session compared to the baseline in both the treatment groups

(rESWT and TENS groups). No prominent and significant differences were identified by analyzing gender and the affected side of the participants in each group.

## Supporting information

**S1 Checklist. CONSORT 2010 checklist of information to include when reporting a randomised trial\*.**
(DOC)

**S1 Data.**
(XLSX)

**S1 Fig.**
(DOC)

**S1 File. Study protocol.**
(DOCX)

## Acknowledgments

We would like to thank all the participants of this study for their valuable contributions.

## Author Contributions

**Conceptualization:** Iresha Dilhari Senarath.

**Data curation:** Iresha Dilhari Senarath, Manoji Pathirage.

**Formal analysis:** Iresha Dilhari Senarath, Randika Dinesh Thalwathte.

**Investigation:** Manoji Pathirage, Senanayake A. M. Kularatne.

**Methodology:** Iresha Dilhari Senarath, Randika Dinesh Thalwathte, Manoji Pathirage, Senanayake A. M. Kularatne.

**Project administration:** Iresha Dilhari Senarath, Manoji Pathirage, Senanayake A. M. Kularatne.

**Resources:** Iresha Dilhari Senarath, Manoji Pathirage, Senanayake A. M. Kularatne.

**Software:** Iresha Dilhari Senarath, Randika Dinesh Thalwathte.

**Supervision:** Manoji Pathirage, Senanayake A. M. Kularatne.

**Validation:** Iresha Dilhari Senarath, Manoji Pathirage, Senanayake A. M. Kularatne.

**Visualization:** Manoji Pathirage.

**Writing – original draft:** Iresha Dilhari Senarath.

**Writing – review & editing:** Manoji Pathirage, Senanayake A. M. Kularatne.

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
