## [Editor Report · Decision Letter 0]

30 May 2022

PONE-D-22-14677The effectiveness of radial Extracorporeal Shock Wave Therapy vs Transcutaneous Electrical Nerve Stimulation in the management of Upper Limb spasticity in chronic-post stroke hemiplegia – A randomized controlled trialPLOS ONE

Dear Dr. Senarath,

Thank you for submitting your manuscript to PLOS ONE. After careful consideration, we feel that it has merit but does not fully meet PLOS ONE’s publication criteria as it currently stands. Therefore, we invite you to submit a revised version of the manuscript that addresses the points raised during the review process. I suggest some corrections and revisions:

1. The authors should improve the english grammatical. 

2. The authors should describe the method of randomization, do they use computerized block randomization? How the conceal the treatment allocation ?

3. The authors should add baseline characteristic table. It is very important to make sure the baseline characteristics similar. 

4. It is better to add the time horizon from stroke onset to subjects recruitment. 

5. It is better to concise in introduction and give more detail in method.

6. Please add the limitation of your study. 

7. It is better to give more detail about baseline NIHSS, mRS, BI, co-morbidities, and co-medication.  Please submit your revised manuscript by Jul 14 2022 11:59PM. If you will need more time than this to complete your revisions, please reply to this message or contact the journal office at plosone@plos.org. Please include the following items when submitting your revised manuscript:A rebuttal letter that responds to each point raised by the academic editor and reviewer(s). You should upload this letter as a separate file labeled 'Response to Reviewers'.A marked-up copy of your manuscript that highlights changes made to the original version. You should upload this as a separate file labeled 'Revised Manuscript with Track Changes'.An unmarked version of your revised paper without tracked changes. You should upload this as a separate file labeled 'Manuscript'.

We look forward to receiving your revised manuscript.

Kind regards,

Rizaldy Taslim Pinzon

Academic Editor

PLOS ONE

Journal Requirements:

4. Please ensure that you refer to Figure 1 in your text as, if accepted, production will need this reference to link the reader to the figure.

Additional Editor Comments:

I suggest some corrections and revisions:

1. The authors should improve the english grammatical.

2. The authors should describe the method of randomization, do they use computerized block randomization? How the conceal the treatment allocation ?

3. The authors should add baseline characteristic table. It is very important to make sure the baseline characteristics similar.

4. It is better to add the time horizon from stroke onset to subjects recruitment.

5. It is better to concise in introduction and give more detail in method.

6. Please add the limitation of your study.

7. It is better to give more detail about baseline NIHSS, mRS, BI, co-morbidities, and co-medication.
---

## [Author Response · Author response to Decision Letter 0]

10 Jun 2022

1. The authors should improve the English grammatical

• Thank you for the comment. After addressing all the other comments, we corrected the grammatical errors by a language expert. (Throughout the manuscript)

2. The authors should describe the method of randomization, do they use computerized block randomization? How do they conceal the treatment allocation?

• The randomization process was done in different steps. It is further described under ‘Methodology-Randomization’. We added more details to the randomization process and treatment allocation. (Page 6)

3. The authors should add a baseline characteristic table. It is very important to make sure the baseline characteristics similar

• A baseline characteristic table is added under the Results section. (Page 9, Table 1)

4. It is better to add the time horizon from stroke onset to subjects’ recruitment

• We described patient recruitment under a new topic as ‘Methodology-Study Design and Patient Recruitment’.

• It describes the time period of patient allocation and stroke onset. (Page 4)

5. It is better to concise in introduction and give more detail in method

• Reduce the content of the introduction and add more details in the Methods section.

• We added study design and patient recruitment under the Methodology section and further described the randomization process. (Page 3, 4, 6)

6. Please add the limitation of your study

• A new paragraph of ‘Limitations’ is included in the manuscript (Page 17)

7. It is better to give more detail about baseline NIHSS, mRS, BI, co-morbidities, and co-medication

• Mainly the assessment was conducted to assess spasticity level (MAS), voluntary control of the upper limb (VCG), and hand functions (FMA-UL and ARAT). Baseline NIHSS, mRS, BI were not taken into the account. They were assessed when the patient was admitted to the hospital as an acute patient. 

• The details about co-morbidities are included under the ‘Results’ section. 

• The details on the medications that are used for co-morbidities are also included in the data sheet.

• A statement is included in the result section stating that ‘These included patients were taking medications only for their present co-morbidities. They were not under any medications to treat spasticity at the stage of patient recruitment and they didn’t take any anti-spastic drugs throughout the treatment protocol. They were only treated with either rESWT or TENS for spasticity management’. (Page 9)

---

## [Decision Letter · Decision Letter 1]

27 Jun 2022

PONE-D-22-14677R1The effectiveness of radial Extracorporeal Shock Wave Therapy vs Transcutaneous Electrical Nerve Stimulation in the management of Upper Limb spasticity in chronic-post stroke hemiplegia – A randomized controlled trialPLOS ONE

Dear Dr. Dilhari Senarath

Thank you for submitting your manuscript to PLOS ONE. After careful consideration, we feel that it has merit but does not fully meet PLOS ONE’s publication criteria as it currently stands. Therefore, we invite you to submit a revised version of the manuscript that addresses the points raised during the review process. Please follow some substantial input and comments from the reviewer. There are some comments and revision in method and the result section. 

We look forward to receiving your revised manuscript.

Kind regards,

Rizaldy Taslim Pinzon

Academic Editor

PLOS ONE

Reviewers' comments:

Reviewer's Responses to Questions

**Comments to the Author**

1. If the authors have adequately addressed your comments raised in a previous round of review and you feel that this manuscript is now acceptable for publication, you may indicate that here to bypass the “Comments to the Author” section, enter your conflict of interest statement in the “Confidential to Editor” section, and submit your "Accept" recommendation.

Reviewer #1: (No Response)

2. Is the manuscript technically sound, and do the data support the conclusions?

Reviewer #1: No

3. Has the statistical analysis been performed appropriately and rigorously? 

Reviewer #1: (No Response)

4. Have the authors made all data underlying the findings in their manuscript fully available?

Reviewer #1: Yes

5. Is the manuscript presented in an intelligible fashion and written in standard English?

Reviewer #1: Yes

6. Review Comments to the Author

Reviewer #1: This study aims to compare the effectiveness of rESW (radial Extracorporeal Shock Wave Therapy) and TENS (Transcutaneous Electrical Nerve Stimulation) in patients with upper limb spasticity in chronic-post stroke hemiplegia. The study used a prospective single-blinded randomized controlled trial to compare the effectiveness of the two treatments in terms of spasticity level on the modified Ashworth scale (MAS), voluntary control of the upper limb (VCG), and hand functions (FMA-UL and ARAT). In this study, the patients were recruited using registry data. The treatments were allocated using a block-randomization and patients from different stroke groups were randomized separately. The null hypothesis and alternative hypothesis were clearly spelled out. Overall, the study design is sound and reasonable. The patient characteristics at baseline were summarized and clearly presented.

However, I have several concerns:

Major concerns:

1. The main conclusion from this study “The rESWT group has shown immediate after-effects at the end of the modality’s first application in spasticity reduction and improving voluntary control of the upper limb but not by the TENS” was not supported by the data. The statistical analyses of the mean differences clearly showed significant differences between T1 & T0 and T2& T0 in both treatment groups.

2. The analyses of median differences showed no significant differences for TENS group in either T1 & T0 or T2& T0 groups. However, as the MAS is an ordinal variable with a 6-point scale, the median analysis is not appropriate and conclusive here. Mean differences are much more informative than median differences here, therefore the conclusion should be based on the analysis of mean differences.

3. This is a comparative study, direct comparisons of the change in MAS (or VCG) between the two treatment groups at the T1 and T2 stage will be more informative.

Minor concerns:

1. The power justification is confusing. Please clarify whether the power calculation is based on the t-test or Wilcoxon-Mann-Whitney test

7. PLOS authors have the option to publish the peer review history of their article (what does this mean?). If published, this will include your full peer review and any attached files.

Reviewer #1: No

---

## [Author Response · Author response to Decision Letter 1]

14 Jul 2022

1. If the authors have adequately addressed your comments raised in a previous round of review and you feel that this manuscript is now acceptable for publication, you may indicate that here to bypass the “Comments to the Author” section, enter your conflict of interest statement in the “Confidential to Editor” section, and submit your "Accept" recommendation.

• Reviewer’s response: No response

• Authors’ response: Not applicable

2. Is the manuscript technically sound, and do the data support the conclusions?

Reviewer’s response: No 

• Authors’ response: 

Thank you for the comment. We have revised our conclusions based on the results that we have obtained.

We included effect sizes to interpret the results in the abstract and the results section.

3. Has the statistical analysis been performed appropriately and rigorously?

• Reviewer’s response: No response

• Authors’ response: Not applicable

4. Have the authors made all data underlying the findings in their manuscript fully available?

• Reviewer’s response: Yes

• Authors’ response: Thank you for the response

5. Is the manuscript presented in an intelligible fashion and written in Standard English?

• Reviewer’s response: Yes

• Authors’ response: Thank you for the response

 

Review Comments to the Author; 

6. Major concerns

• The analyses of median differences showed no significant differences for TENS group in either T1 & T0 or T2& T0 groups. However, as the MAS is an ordinal variable with a 6-point scale, the median analysis is not appropriate and conclusive here. Mean differences are much more informative than median differences here, therefore the conclusion should be based on the analysis of mean differences.

Authors’ response:

As the reviewers were suggested, we conducted the analysis of MAS and VCG with mean values, mean difference%, and paired sample t-test.

The results obtained were incorporated in the abstract, results, and discussion sections of the manuscript.

• The main conclusion from this study “The rESWT group has shown immediate after-effects at the end of the modality’s first application in spasticity reduction and improving voluntary control of the upper limb but not by the TENS” was not supported by the data. The statistical analyses of the mean differences clearly showed significant differences between T1 & T0 and T2& T0 in both treatment groups.

Authors’ response:

According to the above analysis, the statement should be changed as ‘Both groups have shown immediate after-effects at the end of the modality’s first application in spasticity reduction and improving voluntary control of the upper limb but the rESWT group has shown larger improvement compared with TENS.

• This is a comparative study, direct comparisons of the change in MAS (or VCG) between the two treatment groups at the T1 and T2 stage will be more informative.

Authors’ response

As suggested, direct comparisons of MAS and VCG between rESWT and TENS were conducted. It is incorporated in the abstract and results section.

Minor concerns

• The power justification is confusing. Please clarify whether the power calculation is based on the t-test or Wilcoxon-Mann-Whitney test

Authors’ response

In G*power, we used t-tests as the Test family and

Wilcoxon-Mann-Whitney test as the Statistical test for sample size calculation. It was adopted from previous research conducted on rESWT on spasticity reduction. (Page 5)

---

## [Editor Report · Decision Letter 2]

23 Aug 2022

PONE-D-22-14677R2The effectiveness of radial Extracorporeal Shock Wave Therapy vs Transcutaneous Electrical Nerve Stimulation in the management of Upper Limb spasticity in chronic-post stroke hemiplegia – A randomized controlled trialPLOS ONE

Dear Dr. Senarath,

Thank you for submitting your manuscript to PLOS ONE. After careful consideration, we feel that it has merit but does not fully meet PLOS ONE’s publication criteria as it currently stands. Therefore, we invite you to submit a revised version of the manuscript that addresses the points raised during the review process.

 The authors have made several changes. 

Some major concern about the number of co-morbidities.

The total number of subjects is not match with the description of every categories. 

The authors have made changes in the tables, but it is still too complicated. Please revise the limitations of your study.  Please submit your revised manuscript by Oct 07 2022 11:59PM.  If you will need more time than this to complete your revisions, please reply to this message or contact the journal office at plosone@plos.org. Please include the following items when submitting your revised manuscript:A rebuttal letter that responds to each point raised by the academic editor and reviewer(s). You should upload this letter as a separate file labeled 'Response to Reviewers'.A marked-up copy of your manuscript that highlights changes made to the original version. You should upload this as a separate file labeled 'Revised Manuscript with Track Changes'.An unmarked version of your revised paper without tracked changes. You should upload this as a separate file labeled 'Manuscript'.

We look forward to receiving your revised manuscript.

Kind regards,

Rizaldy Taslim Pinzon

Academic Editor

PLOS ONE

Additional Editor Comments:

Thanks for the authors for the appropriate changes.

The authors have made several changes.

Some major concern about the number of co-morbidities.

The total number of subjects is not match with the description of every categories.

The authors have made changes in the tables, but it is still too complicated.
---

## [Author Response · Author response to Decision Letter 2]

5 Sep 2022

1. Thanks for the authors for the appropriate changes.

The authors have made several changes.

• Authors’ response: Thank you for the kind response.

2. Some major concern about the number of co-morbidities.

• Authors’ response: Page 9, re-arranged the paragraph regarding medical co-morbidities.

• Patients 64/106 (60.4%) were affected by other medical co-morbidities, and some may have reported having a single disease and some may have multiple diseases. 

• The patients who is having co-morbidities have taken as 100% and further explained on their disease diagnosis. 53/64 (82.81%) HTN, 40/64 (62.5%) DM, 12/64 (18.75%) IHD, 3/64 (4.68%) CKD, 1/64 (1.56%) AF, 1/64 (1.56%) DLP were reported as a whole. 

• Among those, 16/64 (25%) had only HTN, 7/64 (10.93%), had only DM, 11/64 (17.18%) had HTN and DM, 10/64 (15.62%) had HTN, DM, DLP, and 5/64 (7.81%) had HTN, DM, IHD. 

3. The total number of subjects is not match with the description of every categories.

• Authors’ response: Re-checked the total number of patients written in each category. The total number for each category (variable) is 106.

Treatment option (106)

• rESWT - 53

• TENS – 53

Experimental (rESWT) and control (TENS) groups were further divided according to the stroke type, gender, and affected side. 

Stroke type (106)

• rESWT ischemic stroke – 37

• rESWT hemorrhagic stroke – 16

• TENS ischemic stroke – 37

• TENS hemorrhagic stroke – 16

Gender

• rESWT male – 28

• rESWT female – 25

• TENS male – 19

• TENS female - 34

Affected side

• rESWT right side – 35

• rESWT left side – 18

• TENS right side – 33

• TENS left side - 20

4. The authors have made changes in the tables, but it is still too complicated. 

• Authors’ response: More columns were added to the tables explaining the results and it is revised to display in a simple manner.

5. Please revise the limitations of your study.

• Authors’ response: Thank you for the kind comment. The limitations of the study paragraph are revised accordingly.

---

## [Editor Report · Decision Letter 3]

21 Sep 2022

PONE-D-22-14677R3The effectiveness of radial Extracorporeal Shock Wave Therapy vs Transcutaneous Electrical Nerve Stimulation in the management of Upper Limb spasticity in chronic-post stroke hemiplegia – A randomized controlled trialPLOS ONE

Dear Dr. Senarath

Thank you for submitting your manuscript to PLOS ONE. After careful consideration, we feel that it has merit but does not fully meet PLOS ONE’s publication criteria as it currently stands. Therefore, we invite you to submit a revised version of the manuscript that addresses the points raised during the review process. Thank you for the responses. I I still concern about the data in table 3 and table 4. I ask the authors to review the data again. Some mean differences in table 3 and 4 is much higher than the final (T2) visit score. It is not possible to have higher mean differences than the final scores. The limitations of your study should be from the methodological point of view (the numbers of samples, the site of the study, the blinding or objective measurement, the intent to treat analysis, the adjustment of important prognostic factors).  Please submit your revised manuscript by Nov 05 2022 11:59PM. If you will need more time than this to complete your revisions, please reply to this message or contact the journal office at plosone@plos.org. Please include the following items when submitting your revised manuscript:A rebuttal letter that responds to each point raised by the academic editor and reviewer(s). You should upload this letter as a separate file labeled 'Response to Reviewers'.A marked-up copy of your manuscript that highlights changes made to the original version. You should upload this as a separate file labeled 'Revised Manuscript with Track Changes'.An unmarked version of your revised paper without tracked changes. You should upload this as a separate file labeled 'Manuscript'.

We look forward to receiving your revised manuscript.

Kind regards,

Rizaldy Taslim Pinzon

Academic Editor

PLOS ONE

Journal Requirements:

Additional Editor Comments:

Thank you for the responses. I I still concern about the data in table 3 and table 4. I ask the authors to review the data again. The limitations of your study should be from the methodological point of view (the numbers of samples, the site of the study, the blinding or objective measurement, the intent to treat analysis, the adjustment of important prognostic factors).

---

## [Author Response · Author response to Decision Letter 3]

26 Sep 2022

1. Thank you for the responses. I still concern about the data in table 3 and table 4. I ask the authors to review the data again. Some mean differences in table 3 and 4 is much higher than the final (T2) visit score. It is not possible to have higher mean differences than the final scores.

• Thank you the editors and the reviewers for the kind response.

• As suggested, we reviewed the data again and did the statistical analysis again.

• T0-Baseline values

• T1-values just after 1st treatment application

• T2-values at the end of the four-week treatment protocol

• T1 values are higher than the T0 values as the participants benefited after 1st application of the modality. 

• But they achieved more outcome values after applying the same modality for four weeks, which we assessed at the end of the four-week protocol, T2.

• Therefore, T2 values are higher than both T0 and T1 values.

• It is like; T2>T1>T0

• If we compare the results with a single application (T1) to the baseline and four applications (T2) to the baseline;

• After four applications of the modality patients were presented with great results. 

• Therefore, if we compare the time pairs, the mean differences between the time pairs are like;

• T0&T2>T1&T2>T0&T1

• The table 3 and 4 are reporting the results accordingly.

• Yes, it is true, the mean difference of any time pair cannot be higher than the score at T2.

• In the columns of T0, T1, and T2, we indicate the mean values of the subgroups, and in the columns of the time pairs (T0&T1, T0&T2, T1&T2), we presented the percentage of the mean difference.

• Therefore, values in some of the cells of time pairs can be higher than the T2 value, as they are showing a percentage of the mean difference.

2. The limitations of your study should be from the methodological point of view (the numbers of samples, the site of the study, the blinding or objective measurement, the intent to treat analysis, the adjustment of important prognostic factors). 

• Thank you for the comment. We have revised the limitations section accordingly.

---

## [Editor Report · Decision Letter 4]

11 Oct 2022

PONE-D-22-14677R4The effectiveness of radial Extracorporeal Shock Wave Therapy vs Transcutaneous Electrical Nerve Stimulation in the management of Upper Limb spasticity in chronic-post stroke hemiplegia – A randomized controlled trialPLOS ONE

Dear Dr. Senarath,

Thank you for submitting your manuscript to PLOS ONE. After careful consideration, we feel that it has merit but does not fully meet PLOS ONE’s publication criteria as it currently stands. Therefore, we invite you to submit a revised version of the manuscript that addresses the points raised during the review process.

 - Please mention the aim of your study. It is uncommon to state the hypothesis and null hypothesis.- Some mean difference are more than 50% and 100%. For example the T0 is 6, and the T2 is 32.  Can you tell me the mean difference in table.- There should be abbreviations in every table. 

Please submit your revised manuscript by Nov 25 2022 11:59PM. If you will need more time than this to complete your revisions, please reply to this message or contact the journal office at plosone@plos.org. Please include the following items when submitting your revised manuscript:A rebuttal letter that responds to each point raised by the academic editor and reviewer(s). You should upload this letter as a separate file labeled 'Response to Reviewers'.A marked-up copy of your manuscript that highlights changes made to the original version. You should upload this as a separate file labeled 'Revised Manuscript with Track Changes'.An unmarked version of your revised paper without tracked changes. You should upload this as a separate file labeled 'Manuscript'.

We look forward to receiving your revised manuscript.

Kind regards,

Rizaldy Taslim Pinzon

Academic Editor

PLOS ONE

Additional Editor Comments:

Thank you for the response. We will continue the discussion about mean differences.

---

## [Author Response · Author response to Decision Letter 4]

11 Oct 2022

1. Please mention the aim of your study. It is uncommon to state the hypothesis and null hypothesis.

• Thank you for the comment. We remove the hypothesis from the manuscript.

• The aim of the study is already mentioned in the last sentence of the second paragraph of the introduction section.

• Page 3, Line 69-71

2. Some mean difference are more than 50% and 100%. For example the T0 is 6, and the T2 is 32. Can you tell me the mean difference in table?

• Thank you for the comment. 

• The scores of the MAS and VCG assessment tools range from 0-6. Therefore the mean scores of spasticity grading and voluntary control grading at T0, T1, and T2 time points lie between 0-6 only.

• But, in the FMA-UL tool, the score range from 0 to a total of 66, and in the ARAT scale, the score range from 0 to 57.

• Therefore the mean scores of FMA-UL and ARAT at T0, T1, and T2 can lie between 0-66 and 0-57 respectively.

• The mean scores of MAS and VCG at T0, T1, and T2 are mentioned in the first three columns of Tables 2 and 3 respectively.

• The mean scores of FMA-UL and ARAT at T0, T1, and T2 are mentioned in the first three columns of Tables 4 and 5 respectively.

• To evaluate the improvement by the intervention, we took the percentage of mean difference at each time point; T0 & T1, T0 & T2, and T1 & T2.

• Even though the tools have different scales, as we are taking the percentage of mean difference, it will allow the readers to easily compare the results among sub-groups.

3. There should be abbreviations in every table. 

• We add abbreviations to all the tables accordingly.

---

## [Editor Report · Decision Letter 5]

7 Mar 2023

The effectiveness of radial Extracorporeal Shock Wave Therapy vs Transcutaneous Electrical Nerve Stimulation in the management of Upper Limb spasticity in chronic-post stroke hemiplegia – A randomized controlled trial

PONE-D-22-14677R5

Dear Dr. Senarath

We’re pleased to inform you that your manuscript has been judged scientifically suitable for publication and will be formally accepted for publication once it meets all outstanding technical requirements.

Kind regards,

Rizaldy Taslim Pinzon

Academic Editor

PLOS ONE

Additional Editor Comments (optional):

There is novelty in this topic. Thank you for your prompt reply and responses.
---

## [Editor Report · Acceptance letter]

22 Mar 2023

PONE-D-22-14677R5 

The effectiveness of radial Extracorporeal Shock Wave Therapy vs Transcutaneous Electrical Nerve Stimulation in the management of Upper Limb spasticity in chronic-post stroke hemiplegia – A randomized controlled trial 

Dear Dr. Senarath:

I'm pleased to inform you that your manuscript has been deemed suitable for publication in PLOS ONE. Congratulations! Your manuscript is now with our production department. 

Kind regards, 

on behalf of

Dr. Rizaldy Taslim Pinzon 

Academic Editor

PLOS ONE